# GenS: Generalizable Neural Surface Reconstruction from Multi-View Images

**Rui Peng[1,2]  Xiaodong Gu[3]  Luyang Tang[1]  Shihe Shen[1]  Fanqi Yu[1]  Ronggang Wang[✉,1,2]**

[1]School of Electronic and Computer Engineering, Peking University
[2]Peng Cheng Laboratory  [3]Alibaba Group
ruipeng@stu.pku.edu.cn   rgwang@pkusz.edu.cn

## Abstract

Combining the signed distance function (SDF) and differentiable volume rendering has emerged as a powerful paradigm for surface reconstruction from multi-view images without 3D supervision. However, current methods are impeded by requiring long-time per-scene optimizations and cannot generalize to new scenes. In this paper, we present GenS, an end-to-end generalizable neural surface reconstruction model. Unlike coordinate-based methods that train a separate network for each scene, we construct a generalized multi-scale volume to directly encode all scenes. Compared with existing solutions, our representation is more powerful, which can recover high-frequency details while maintaining global smoothness. Meanwhile, we introduce a multi-scale feature-metric consistency to impose the multi-view consistency in a more discriminative multi-scale feature space, which is robust to the failures of the photometric consistency. And the learnable feature can be self-enhanced to continuously improve the matching accuracy and mitigate aggregation ambiguity. Furthermore, we design a view contrast loss to force the model to be robust to those regions covered by few viewpoints through distilling the geometric prior from dense input to sparse input. Extensive experiments on popular benchmarks show that our model can generalize well to new scenes and outperform existing state-of-the-art methods even those employing ground-truth depth supervision. Code will be available at `https://github.com/prstrive/GenS`.

## 1  Introduction

Surface reconstruction from multi-view images is a cornerstone task in computer vision with many applications in virtual reality, autonomous driving, robotics, etc. Typical solutions [15, 16, 7, 43, 58, 56, 10, 36] in the past were mostly based on a multi-step pipeline, which includes depth estimation, depth fusion and meshing. Although they have demonstrated their excellent performance, the procedure is cumbersome and inevitably introduces cumulative errors. While several early works [31, 61] used differentiable surface rendering to directly reconstruct surfaces, recent works [34, 49, 60], inspired by the huge success of neural radiance field (NeRF) [27] in synthesizing novel views, follow the volume rendering [24] to represent the 3D geometry with an occupancy field [25] or signed distance function (SDF) [35] and can achieve more impressive results.

The key idea of these approaches is to train a compact multi-layer perceptrons (MLPs) to predict the implicit representation (e,g., SDF) of each sampled point on camera rays. The density of volume rendering is then regarded as a function of this implicit representation, and alpha-composition of samples is performed to produce the corresponding pixel color and geometry information. However, existing methods are hampered by requiring a lengthy per-scene optimization procedure and cannot generalize to new scenes, which makes them infeasible for many application scenarios. A recent method [23] attempts to address these issues through conditioning the SDF-induced model with

37th Conference on Neural Information Processing Systems (NeurIPS 2023).

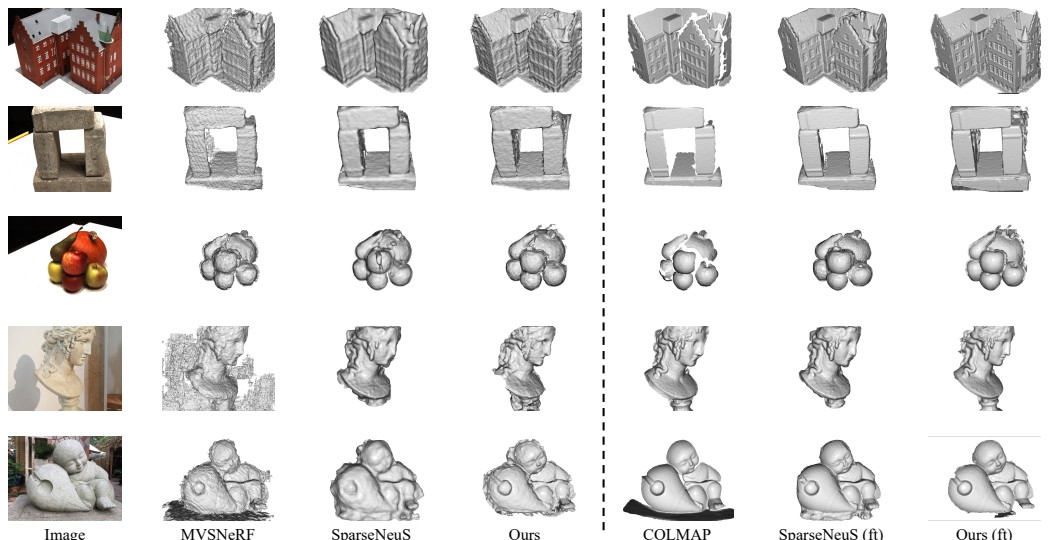

| Image | MVSNeRF | SparseNeuS | Ours | COLMAP | SparseNeuS (ft) | Ours (ft) |

Figure 1: **Qualitative comparisons on DTU and BlendedMVS datasets with sparse inputs.**

features extracted from sparse nearby views. Nevertheless, its accuracy is limited due to the smooth reconstruction, and the multi-stage process it relies on is prone to introducing cumulative errors. In this paper, we seek to establish an end-to-end generalizable model which can efficiently infer finer 3D structure. Compared with existing methods [49, 60], this generalization system faces more challenging problems. First, it's non-trivial to efficiently represent the scene. Previous methods [29, 23, 3, 50] either build a global volume or employ feature projections, but they have proven to be either lacking in detail or unsuitable for view independent surface reconstruction. Second, relying only on the rendering loss is difficult to reconstruct compact geometry, since the multi-view consistency is ignored. And we found that the ordinary photometric consistency also cannot effectively solve this problem for our generalizable model because of the existence of ambiguous areas such as low-texture and reflection. Last but not least, since generalizable models heavily rely on aggregation quality, how to infer smooth geometry when the input is sparse is a thorny issue.

To this end, we introduce GenS to tackle these challenges. The main ideas behind are as follows: 1) We first construct a generalized multi-scale volume to represent the scene, which preserves global smoothness through the low-resolution volumes and recovers geometric details from high-resolution volumes. Meanwhile, low-dimensional features make our model more lightweight compared to a single large-width volume. 2) We introduce the multi-scale feature-metric consistency, which enforces multi-view consistency in the multi-scale feature space, to replace the common photometric consistency. Compared with the original image space, learnable multi-scale features can provide more discriminative representation, and the feature space can be self-enhanced during the generalization training process to continuously improve the matching accuracy. 3) Inspired by the fact that the reconstruction with dense inputs is more accurate, we propose a view contrast loss to force the model to better perceive the geometry of regions visible by few viewpoints through teaching the reconstruction from sparse inputs with dense inputs.

To demonstrate the quantitative and qualitative effectiveness of GenS, we conduct extensive experiments on DTU [12] and BlendedMVS [59] datasets. Results show that our model can outperform existing state-of-the-art generalizable method [23], and even recent method [40] which adopts the ground-truth depth for supervision. Compared with the per-scene overfitting methods [49, 60, 61, 34, 23], we can also achieve comparable or superior results with dense inputs. Some comparisons are shown in Fig. 1. In summary, our main contributions are highlighted below: **a)** We present a powerful representation based on our generalized multi-scale volume, which can efficiently reconstruct smooth and detail surfaces from multi-view images. **b)** We introduce a more discriminative multi-scale feature-metric consistency to successfully constrain the geometry, which helps the generalization model converge to the optimum. **c)** We propose a view contrast loss to improve the geometric smoothness and accuracy when the visible viewpoint is limited. **d)** Our model can be trained end-to-end and achieve state-of-the-art reconstructions in both generic setting and per-scene optimization setting.

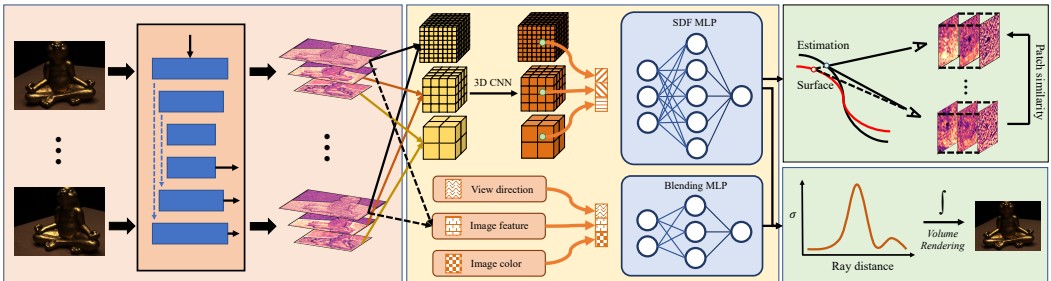

Figure 2: **Illustration of GenS.** We first extract multi-scale features through a FPN network. The generalized multi-scale volume is then reconstructed with the corresponding scale feature. We employ the same blending strategy as [50] to estimate the appearance of each point on a ray, and adopt the volume rendering to recover the color of a pixel. We design the multi-scale feature-metric consistency to constrain the geometry as shown in the top right. For convenience, we omit some losses that will be detailed later.

## 2 Related work

**Classical multi-view reconstruction.** Reconstructing 3D geometry from multi-view images is a longstanding problem in the field of 3D vision. Classical algorithms mainly adopt depth-based or voxel-based methodology to solve this problem. Multi-view stereo (MVS) is a typical class of depth-based methods, which takes stereo correspondence from multiple images as the main cue to reconstruct depth maps. While previous traditional MVS methods [1, 43, 7, 6, 56, 44] relied on the hand-crafted similarity metrics, many recent learning-based methods [58, 10, 48, 36] apply deep learning to achieve more discriminative matching. These methods go through complicated procedures to retrieve surface, including depth filtering, fusion and meshing [15, 2], and are prone to cumulative errors. On the other hand, voxel-based methods [45, 18, 11, 32] directly model objects in a volume, but they are restricted to memory, which is the common drawback of the volumetric representation, and cannot achieve high accuracy.

**Neural surface.** Due to the notable advantages of being able to achieve high spatial resolution, neural implicit functions have recently gained a lot of attention and have emerged as an effective representation of 3D geometry [47, 25, 35, 8, 26, 30, 37, 42] and appearance [20, 27, 21, 33, 46, 62, 38, 28]. Furthermore, some works [27, 22, 31] have proposed to train models without 3D supervision via differentiable rendering, e.g., surface rendering and volume rendering. Methods adopt surface rendering [31, 61, 65] only consider a single surface intersection point for each ray and fail to reconstruct complex objects, and they are restricted by the need of accurate object masks and careful weight initialization. On the contrary, recent methods use volume rendering [34, 49, 60, 64] to take multiple points along the ray into consideration and achieve more impressive results. However, either type of method requires an expensive per-scene optimization and cannot generalize to new scenes.

**Generalizable neural surface.** In the field of novel view synthesis, some methods [3, 50, 63, 13] have successfully introduced the generalization into rendering methods. These methods suffer from the same problem as NeRF: the geometry is ambiguous. Few works have focused on the generalization of neural surface reconstruction. A recent study, SparseNeuS [23], is the first attempt to achieve this by reconstructing the surface from nearby viewpoints in a multi-stage manner. Nevertheless, its reconstruction lacks details, and same to the classical 3D reconstruction, the multi-stage pipeline may accumulates errors at each stage. On the contrary, our designed model can be trained end-to-end and reconstruct smoother and more refined geometries.

## 3 Method

Given $N$ posed images of an object taken from different viewpoints, our goal is to reconstruct the surface as an implicit function without expensive per-scene optimization or only by fast fine-tuning. Our overall framework is depicted in Fig 2. We first introduce how to infer the geometry and appearance from the generalized multi-scale volume in Sec. 3.1, then elaborate on the necessity and implementation of the multi-scale feature-metric consistency in Sec. 3.2, and finally detail the realization of view contrast loss in Sec. 3.3 and the overall pipeline in Sec. 3.4.

## 3.1 Geometry and color reasoning from the generalized multi-scale volume

Compared with existing solution [23], which relies on a single volume and multi-stage strategy, we have three main advantages. First of all, our generalized multi-scale volume is a more powerful representation, which implicitly decouples geometry into base structures in low-resolution volumes and high-frequency details in high-resolution volumes. Second, with the low-dimensional features, we can construct multi-scale volumes with higher resolution and less memory consumption than a single large-width volume. Besides, our model can be trained end-to-end, avoiding cumulative errors.

**Generalized multi-scale volume construction.** Suppose there are $N$ images $\{I_i\}_{i=0}^{N-1}$ of an object, we first apply the FPN network [19] to extract multi-scale feature maps $\{F_i^j\}_{i,j=0,0}^{N-1,L-1}$ for all images with shared weights, and different volumes are then constructed from features at corresponding scales. In this paper, we define a bounding box of interest in the reference frustum like [23] and in the world coordinate system like [49, 66] when dense inputs are available. We adopt a combination of $L$ volumes $\{V_j\}_{j=0}^{L-1}$, which cover the same region but with different resolutions $Ch \times \frac{D}{2^j} \times \frac{H}{2^j} \times \frac{W}{2^j}$.

Here, we discuss at the first scale and omit the scale subscript $j$ for convenience. Given camera intrinsics $\{K_i\}_{i=0}^{N-1}$ and extrinsics $\{[R,t]_i\}_{i=0}^{N-1}$, we first project the voxel $v = (x, y, z)$ onto viewpoint $i$'s pixel position:

$$q_i(v) = \pi(K_i R_i^T (v - t_i)), \tag{1}$$

where $\pi((x, y, z)^T) = (\frac{x}{z}, \frac{y}{z})^T$ is an operator to convert homogeneous coordinates to cartesian coordinates. Then we can get the corresponding feature of each voxel on $i_{th}$ viewpoint through bilinear interpolation $F_i(v) = F_i < q_i(v) >$. To fuse features from all viewpoints $\{F_i(v)\}_{i=0}^{N-1}$, we adopt the same aggregation strategies to generate cost volume as in [50] that concatenates mean and variance to simultaneously capture statistical and semantic information: $B(v) = [Mean(v), Var(v)]$.

Simply repeating the above process on features and volumes of all $L$ scales, we can get the multi-scale cost volumes $\{B_j\}_{j=0}^{L-1}$. Next, we further design an efficient multi-scale 3D network $\psi$ to refine these cost volumes in one forward, starting from the finest volume and injecting the others into different stages of the model to save memory. The output of the 3D network $\{V_j\}_{j=0}^{L-1} = \psi(\{B_j\}_{j=0}^{L-1})$ is the multi-scale volume that we need to infer the geometry.

**Geometry reasoning.** For an arbitrary 3D point $p = (x, y, z)$, we first get the interpolation of volumes at all scales $\{V_j(p)\}_{j=0}^{L-1}$ through trilinear sampling, and then concatenate them as the final feature $\mathcal{F}(p) \in \mathbb{R}^{Ch_1}$, where $Ch_1 = L \times Ch$. Combining the feature and the point position, an MLP network is applied to predict the corresponding SDF value: $sdf_\theta : \mathbb{R}^3 \times \mathbb{R}^{Ch_1} \to \mathbb{R}$. And the surface is represented by the zero-level set of the SDF value:

$$S = \{p \in \mathbb{R}^3 | sdf_\theta(p, \mathcal{F}(p)) = 0\}. \tag{2}$$

**Color prediction.** We refer to the first viewpoint $I_0$ as the reference image. To predict the color of each point on a ray, we employ the blending strategy similar to [50]. We first project the 3D point $p$ to source views' pixel position according to Eq. 1, and interpolate the corresponding colors $\{I_i(p)\}_{i=1}^{N-1}$ and features $\{F_i(p)\}_{i=1}^{N-1}$. Here, we only use the highest resolution features to predict the color. Next, an MLP network take the concatenation of features and viewing direction differences $\Delta d = d - d_i$ as input, to predict the softmax-activated blending weights $\{w_i(p)\}_{i=1}^{N-1}$ of each source view, and the final color is blended as the weighted sum of source colors:

$$c(p) = \sum_{i=1}^{N-1} I_i(p) w_i(p). \tag{3}$$

**SDF-based volume rendering.** Given the density $\{\sigma_i\}_{i=1}^{M}$ and color $\{c_i\}_{i=1}^{M}$ of $M$ samples along the ray $p(t) = o + td$ emitting from camera center $o$ to pixel $q$ in view direction $d$, NeRF [27] approximates the color using numerical quadrature:

$$\hat{C} = \sum_{i=1}^{M} T_i \alpha_i c_i, \; T_i = \prod_{j=1}^{i-1} (1 - \alpha_j), \tag{4}$$

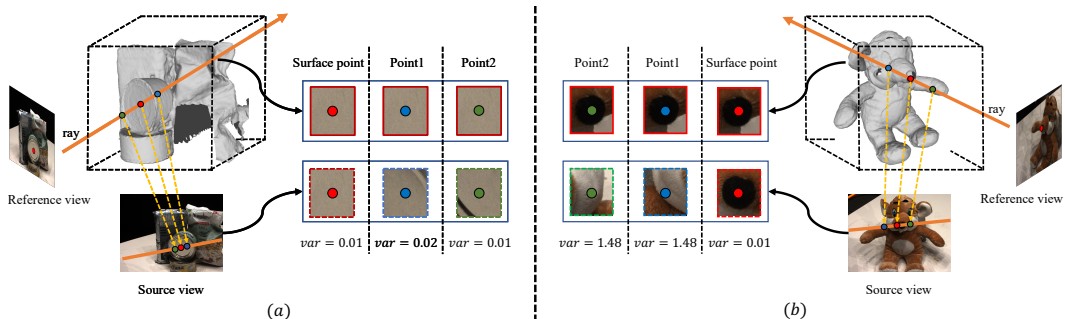

Figure 3: **Multi-view aggregation ambiguity.** Here, we take two viewpoints as an example. (a) For those low-texture regions, sampling points near the surface may get the same aggregation and lack discriminability. (b) The aggregation of points away from the surface are random and hard to infer the accurate geometry, e.g., two sampling points may get the same aggregation even with different SDF value.

where $T_i$ is the accumulated transmittance, and $\alpha_i = 1 - \exp(-\sigma_i \delta_i)$ in original volume rendering. To better approximation the geometry of the scene, NeuS [49] proposed an unbiased and occlusion-aware weighting method to incorporate signed distance, and the $\alpha_i$ is formulated as:

$$\alpha_i = \max\left(\frac{\Phi_s(sdf(p(t_i))) - \Phi_s(sdf(p(t_{i+1})))}{\Phi_s(sdf(p(t_i)))}, 0\right). \tag{5}$$

Here, $\Phi_s(x) = (1 + e^{-sx})^{-1}$ is the sigmoid function and $s$ is an anneal factor. Readers can refer to [49] for more details.

## 3.2 Multi-scale feature-metric consistency

Rendering loss tends to trap the model into sub-optimization since it only considers a single point and ignores the consistency among multiple viewpoints. To mitigate this problem, a straightforward practice is to project the image patches of multiple views to the estimated surface location based on the local planar assumption and rely on the photometric consistency to enforce the multi-view consistency. However, we found this solution works well for per-scene overfitting training [4, 5] but brings limited benefits to generalization training.

We analyze that the main reason may be the failure of photometric consistency, which becomes more challenging for generalization training. As proven in recent self-supervised multi-view stereo methods [55, 54, 57, 39], the assumption of photometric consistency isn't always effective, and the predicted geometry still has significant holes even in combination with the robust patch similarity like SSIM [52]. As the coordinate-based methods train models separately for each scene to directly overfit the scene, they have greater potential to converge to the optimum. However, our generalization model encodes all scenes with one model, and it requires image features to infer geometry, which makes the model rely heavily on the discriminability of features, e.g., regions like low-texture and reflection become more critical for degrading results. As shown in Fig 3 (a), those regions violating photometric consistency not only reduce

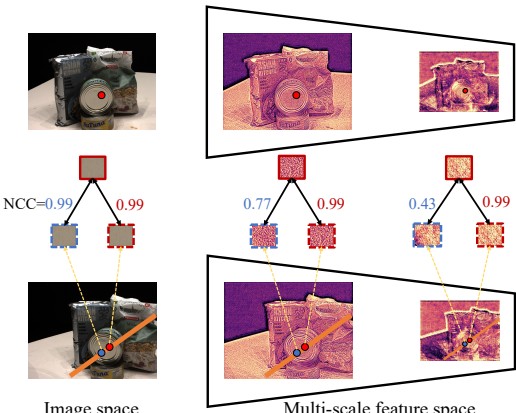

Figure 4: **Multi-scale feature space.** The feature space is more discriminative than ordinary image space, and is more potential to find the corresponding point during matching.

the accuracy of multi-view matching, but also decrease the discriminability of generalization model's input (we call this aggregation ambiguity), while the input of overfitting methods are distinct (3D coordinate).

To overcome these challenges, we propose the multi-scale feature-metric consistency to measure the consistency between views in a multi-scale feature space, as shown in Fig. 4. There are three main advantages of doing this way. First of all, the learnable feature is proven to be more discriminative than the original image [14], especially on those ambiguous regions like low-texture and reflection. Second, due to the larger receptive field, multi-scale information is conducive to improving the matching accuracy, and allows the model to be assisted by global information while recovering details. More importantly, the feature discriminability can be continuously self-enhanced in the process of generalization training. The multi-scale feature space can train a powerful model through more accurate matching, and the more powerful model can in turn lead to a more discriminative feature space. And the enhanced feature can further mitigate the aforementioned aggregation ambiguity. These advantages have been proven in Tab. 3.

To generate the geometry, we adopt the same approximate method as [5] to directly locate the zero-level set of the SDF. As shown in Fig. 5, We first find the interval where a ray intersects the surface by checking whether the signs of the SDFs of two adjacent sampling points are different. To handle occlusion, we only extract the surface within the first interval. Suppose the two samples of the interval are $p_1$ and $p_2$, and their distances to the camera center are $t_1$ and $t_2$ respectively, our goal is to compute the position of $p_s$. Here, we rely on an assumption that two adjacent samples are close enough that the near surface can be regarded as a local plane. In this way, we can get two similar triangles:

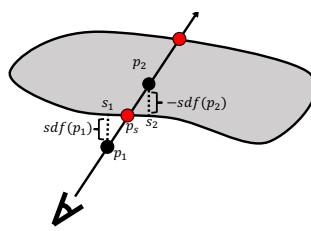

Figure 5: **Locating the surface of a ray.**

$$\triangle p_2 p_s s_2 \backsim \triangle p_1 p_s s_1, \qquad (6)$$

Therefore, we can approximate the distance from the surface to the camera center $t_s$ as:

$$\frac{-sdf(p_2)}{sdf(p_1)} = \frac{t_2 - t_s}{t_s - t_1} \Rightarrow t_s = \frac{sdf(p_1)t_2 - sdf(p_2)t_1}{t_1 - t_2}. \qquad (7)$$

We thus can get the coordinate of the surface point $p_s = o + t_s d$.

Through the automatic differentiation of the SDF network at $p_s$, we can get the corresponding normal $n_s$. Based on the assumption that the local surface centered at $p_s$ is a plane of normal $n_s$, we can find the corresponding pixel position $q_i$ in $i_{th}$ source view that correspond to the pixel $q_0$ in reference view:

$$q_i = H_i q_0, \ H_i = K_i (R_i R_0^T + \frac{R_i(R_i^T t_i - R_0^T t_0)n_s^T}{n_s^T p_s})K_0^{-1}. \qquad (8)$$

For a pixel patch $\mathbf{q}_0$ in the reference view, we can find the corresponding source patch through passing all pixels to Eq. 8 like $\mathbf{q}_i = H_i \mathbf{q}_0$. Regardless of occlusion, if the estimated surface $p_s$ is accurate, then these corresponding patches should also be consistent. In this paper, we measure patch consistency in a multi-scale feature space. We only apply features at the top 3 scales, since features at lower scales lose a lot of structural information. Therefore, for a pixel patch at a certain view, we can get the multi-scale patches $\{F_j < \mathbf{q} >\}_{j=0}^2$ through bilinear interpolation, and we upsample and concatenate them together as input $F'$, whose channel is $Ch_2 = 3 \times Ch$, for patch similarity measure. Here, we employ the normalization cross correlation (NCC) to compute the feature-space consistency:

$$NCC_i = \frac{1}{Ch_2} \sum_{l=0}^{Ch_2-1} \frac{Cov(F'_{0l}, F'_{il})}{\sqrt{Var(F'_{0l})Var(F'_{il})}}, \qquad (9)$$

where $Cov$ denotes covariance and $Var$ refers to variance. Following the common solution in multi-view stereo field [7], we compute the final multi-scale feature-space consistency loss as the average of the best $K$ NCCs:

$$L_{mfc} = \frac{1}{K} \sum_{k=0}^{K-1} (1 - NCC_k). \qquad (10)$$

### 3.3 View contrast loss

For a 3D structure captured by multiple viewpoints, there is a fact that some regions are covered by enough viewpoints, while some regions are only visible to a few viewpoints. Compared with the

former, the aggregated features of the latter are more likely to be polluted by irrelevant rays, making them less predictable. To solve this problem, we design a view contrast loss to improve the accuracy of the reconstruction when visible views are limited, which enforces the geometric estimation to be the same under different inputs of the same scene.

We empirically lets results from dense inputs to supervise results of sparse inputs. Specially, taking a set of multi-view images as input, we first reconstruct a multi-scale volume as a teacher, which is used to infer the finer SDF value $s$ for a set of 3D points $P$. Then we build a student multi-scale volume from sparse input views and estimate the corresponding SDF value $s'$. Meanwhile, as shown in Fig. 3 (b), we found that only the sampling points falling on the surface have positive epipolar correspondences, and their aggregated features are more meaningful, while other samples are more random, and may obtain the same aggregation even if their SDF values are different. As shown in Fig. 6, we thus only calculate the consistency loss for near-surface points, whose finer SDF values are more accurate:

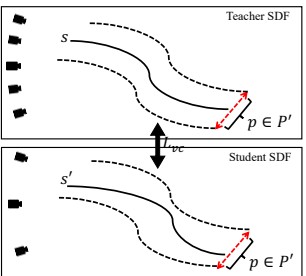

Figure 6: **Visualization of view contrast loss.**

$$L_{vc} = \frac{1}{|P'|} \sum_{p \in P'} |s(p) - s'(p)|, \tag{11}$$

where $P'$ is a set of points close to the surface inferred from the fine SDF according to Eq. 7.

### 3.4 Overall pipeline

This section will introduce some implementation details and crucial components of our model including generalization training and fine-tuning.

**Loss function.** The overall loss function is defined as:

$$L = L_{color} + \alpha L_{mfc} + \beta L_{vc} + L_{reg}. \tag{12}$$

For a batch of sampled pixel set $Q$, the color loss is computed as the L1 distance between the rendered color and the ground-truth:

$$L_{color} = \frac{1}{|Q|} \sum_{q \in Q} |C(q) - \hat{C}(q)|. \tag{13}$$

To make the geometry more compact and accurate, we apply the regularization loss which is composed of four terms:

$$L_{reg} = \gamma L_{ek} + \eta L_{smooth} + \lambda L_{tv} + \delta L_{sparse}. \tag{14}$$

Eikonal loss [9] is employed to regularize SDF values of all sampled points $P$:

$$L_{ek} = \frac{1}{|P|} \sum_{p \in P} (||\nabla sdf(p)||_2 - 1)^2. \tag{15}$$

To maintain the smooth of the surface, we introduce a regularization to the gradient of the normal:

$$L_{smooth} = \frac{1}{|Q|} \sum_{q \in Q} ||n_{grad}(q)||_2, \tag{16}$$

where $n_{grad}(q)$ is the alpha composition of normal gradient $\nabla^2 sdf(q)$ in a ray through pixel $q$. Besides, we also adopt the total variation (TV) regularization [41] for our multi-scale volumes:

$$L_{tv} = \sum_{j=0}^{L-1} \sqrt{\Delta_x^2(V_j) + \Delta_y^2(V_j) + \Delta_z^2(V_j)}. \tag{17}$$

To clean the geometric estimation, we introduce a sparsity prior:

$$L_{sparse} = \frac{1}{|P|} \sum_{p \in P} \exp(-\tau |sdf(p)|). \tag{18}$$

| Method | 24 | 37 | 40 | 55 | 63 | 65 | 69 | 83 | 97 | 105 | 106 | 110 | 114 | 118 | 122 | Mean |
|---|---|---|---|---|---|---|---|---|---|---|---|---|---|---|---|---|
| VolRecon* [40] | 1.20 | 2.59 | 1.56 | 1.08 | 1.43 | 1.92 | 1.11 | 1.48 | 1.42 | 1.05 | 1.19 | 1.38 | 0.74 | 1.23 | 1.27 | 1.38 |
| PixelNerf [63] | 5.13 | 8.07 | 5.85 | 4.40 | 7.11 | 4.64 | 5.68 | 6.76 | 9.05 | 6.11 | 3.95 | 5.92 | 6.26 | 6.89 | 6.93 | 6.28 |
| IBRNet [50] | 2.29 | 3.70 | 2.66 | 1.83 | 3.02 | 2.83 | 1.77 | 2.28 | 2.73 | 1.96 | 1.87 | 2.13 | 1.58 | 2.05 | 2.09 | 2.32 |
| MVSNerf [3] | 1.96 | 3.27 | 2.54 | 1.93 | 2.57 | 2.71 | 1.82 | 1.72 | 2.29 | 1.75 | 1.72 | 1.47 | 1.29 | 2.09 | 2.26 | 2.09 |
| SparseNeuS [23] | 1.68 | 3.06 | 2.25 | 1.10 | 2.37 | 2.18 | 1.28 | **1.47** | 1.80 | 1.23 | 1.19 | 1.17 | 0.75 | 1.56 | 1.55 | 1.64 |
| **GenS** | **1.45** | **2.77** | **1.69** | **0.97** | **1.54** | **1.90** | **1.03** | 1.49 | **1.36** | **0.97** | **1.07** | **0.97** | **0.62** | **1.14** | **1.16** | **1.34** |
| NeuS [49] | 4.57 | 4.49 | 3.97 | 4.32 | 4.63 | 1.95 | 4.68 | 3.83 | 4.15 | 2.50 | 1.52 | 6.47 | 1.26 | 5.57 | 6.11 | 4.00 |
| VolSDF [60] | 4.03 | 4.21 | 6.12 | 0.91 | 8.24 | 1.73 | 2.74 | 1.82 | 5.14 | 3.09 | 2.08 | 4.81 | 0.60 | 3.51 | 2.18 | 3.41 |
| IBRNet (ft) | 1.67 | 2.97 | 2.26 | 1.56 | 2.52 | 2.30 | 1.50 | 2.05 | 2.02 | 1.73 | 1.66 | 1.63 | 1.17 | 1.84 | 1.61 | 1.90 |
| COLMAP [44] | **0.90** | 2.89 | 1.63 | 1.08 | 2.18 | 1.94 | 1.61 | 1.30 | 2.34 | 1.28 | 1.10 | 1.42 | 0.76 | 1.17 | 1.14 | 1.52 |
| SparseNeuS (ft) | 1.29 | **2.27** | 1.57 | 0.88 | 1.61 | 1.86 | 1.06 | 1.27 | 1.42 | 1.07 | 0.99 | 0.87 | 0.54 | 1.15 | 1.18 | 1.27 |
| **GenS (ft)** | 0.91 | 2.33 | **1.46** | **0.75** | **1.02** | **1.58** | **0.74** | **1.16** | **1.05** | **0.77** | **0.88** | **0.56** | **0.49** | **0.78** | **0.93** | **1.03** |

Table 1: **Quantitative results of Chamfer Distance on DTU dataset with sparse inputs.** '*' denotes that the method needs the ground-truth depth for supervision.

**Generalization training.** We select $N = 4$ for sparse setting and $N = 19$ for dense setting. We use Adam optimizer [17] with the base learning rate of 1e-3 for feature network and 5e-4 for other MLPs. We train the joint loss for 16 epochs on two A100 GPUs. We increase the value of $\alpha$ from 0 to 1 and in the first 2 epochs. In our implementation, we generate the surface points $P'$ of each image of the model trained with dense input first, and then distill the model with sparse input, with $\beta$ set to 1. We build the generalized multi-scale volume with 5 scales, whose resolution increase from $2^4$ to $2^8$. Each volume is equipped with thin features with only 4 feature channels, which allows us to save memory compared to general single volume methods.

**Fine-tuning.** After generalization training, we first reconstruct the generalized multi-scale volume, which has encoded the geometry information. Then we sparse the multi-scale volume by pruning voxels far from the surface. During fine-tuning, we abandon the feature network, and directly optimize the multi-scale volume and MLPs. With the generalization prior, we can achieve state-of-the-art performance in only about 20 minutes of fine-tuning.

## 4 Experiments

We demonstrate the state-of-the-art performance of GenS with comprehensive experiments and verify the effectiveness of each module through ablation studies. We first introduce the datasets and then analyze our results.

**Datasets.** We conduct experiments on both DTU [12] and BlendedMVS [59] datasets as previous methods [49, 60, 23]. Our generalization model is trained on DTU dataset, which is an indoor MVS dataset with 124 different scenes scaned from 49 or 64 views with fixed camera trajectories. Following [61, 49, 23], we take the same 15 scenes for testing. The training set is defined as in [58, 36], and the test scenes contained therein are removed. We also evaluate our model on BlendedMVS, which is a large-scale synthetic dataset. Each scene is scaned from different number of views, and all images has a resolution of $768 \times 576$. We report the Chamfer Distance for DTU, and show some visual effects for BlendedMVS.

### 4.1 Comparisons

**Results on DTU.** We first adopt the same testing split and configuration as [23] to compare with existing generalizable methods [3, 50, 23, 63]. The results shown in Tab. 1 indicate that our model outperforms existing methods by a significant margin, and this advantage can be amplified after rapid fine-tuning (about 20 mins). Even compared with recent method [40], which adopts the ground-truth depth for supervision, our model can achieve superior results. The qualitative results in Fig. 1 show that our reconstruction exhibits finer details. We further conducted more experiments on DTU to compare with per-scene overfitting methods with more input views. The quantitative comparisons in Tab. 2 show that our model can surpass some methods [61, 34, 49, 60] just through a very fast network inference, i.e., we can achieve more than 34% improvement on scene 24 compared with [49]. After a fast fine-tuning, the performance can be significantly improved, and even surpassing recent SOTA works [4, 5, 51]. Some visualization results in Fig. 7 depict that our model trained on large amounts of data is more robust to ambiguous regions.

| Method | 24 | 37 | 40 | 55 | 63 | 65 | 69 | 83 | 97 | 105 | 106 | 110 | 114 | 118 | 122 | Mean |
|---|---|---|---|---|---|---|---|---|---|---|---|---|---|---|---|---|
| IDR [61] | 1.63 | 1.87 | 0.63 | 0.48 | 1.04 | 0.79 | 0.77 | 1.33 | 1.16 | 0.76 | 0.67 | 0.90 | 0.42 | 0.51 | 0.53 | 0.90 |
| MVSDF [65] | 0.83 | 1.76 | 0.88 | 0.44 | 1.11 | 0.90 | 0.75 | 1.26 | 1.02 | 1.35 | 0.87 | 0.84 | 0.34 | 0.47 | 0.46 | 0.88 |
| COLMAP [44] | **0.45** | 0.91 | 0.37 | **0.37** | 0.90 | 1.00 | 0.54 | 1.22 | 1.08 | **0.64** | **0.48** | **0.59** | 0.32 | 0.45 | 0.43 | 0.65 |
| NeRF [27] | 1.90 | 1.60 | 1.85 | 0.58 | 2.28 | 1.27 | 1.47 | 1.67 | 2.05 | 1.07 | 0.88 | 2.53 | 1.06 | 1.15 | 0.96 | 1.49 |
| UNISURF [34] | 1.32 | 1.36 | 1.72 | 0.44 | 1.35 | 0.79 | 0.80 | 1.47 | 1.37 | 0.89 | 0.59 | 1.47 | 0.46 | 0.59 | 0.62 | 1.02 |
| VolSDF [60] | 1.14 | 1.26 | 0.81 | 0.49 | 1.25 | 0.70 | 0.72 | 1.29 | 1.18 | 0.70 | 0.66 | 1.08 | 0.42 | 0.61 | 0.55 | 0.86 |
| NeuS [49] | 1.00 | 1.37 | 0.93 | 0.43 | 1.10 | 0.65 | 0.57 | 1.48 | 1.09 | 0.83 | 0.52 | 1.20 | 0.35 | 0.49 | 0.54 | 0.84 |
| HF-NeuS [51] | 0.76 | 1.32 | 0.70 | 0.39 | 1.06 | 0.63 | 0.63 | **1.15** | 1.12 | 0.80 | 0.52 | 1.22 | 0.33 | 0.49 | 0.50 | 0.77 |
| Voxurf [53] | 0.65 | 0.74 | 0.39 | **0.35** | 0.96 | 0.64 | 0.85 | 1.58 | 1.01 | 0.68 | 0.60 | 1.11 | 0.37 | 0.45 | 0.47 | 0.72 |
| NeuralWarp [4] | 0.49 | **0.71** | 0.38 | 0.38 | **0.79** | 0.81 | 0.82 | 1.20 | 1.06 | 0.68 | 0.66 | 0.74 | 0.41 | 0.63 | 0.51 | 0.68 |
| Geo-NeuS* [5] | 0.46 | 0.83 | 0.38 | 0.39 | 0.88 | **0.61** | **0.51** | 1.26 | **0.92** | 0.68 | 0.57 | 0.82 | **0.30** | **0.41** | **0.42** | 0.63 |
| **GenS** | 0.66 | 1.01 | 0.71 | 0.43 | 1.06 | 0.99 | 0.73 | 1.43 | 1.18 | 0.78 | 0.64 | 0.93 | 0.38 | 0.54 | 0.54 | 0.80 |
| **GenS (ft)** | 0.55 | **0.71** | 0.39 | 0.38 | **0.79** | 0.65 | 0.57 | 1.29 | 0.96 | **0.64** | 0.49 | **0.59** | 0.33 | 0.44 | 0.45 | **0.62** |

Table 2: **Quantitative results of Chamfer Distance on DTU dataset with dense inputs.** '*' denotes that we retrain the Geo-NeuS without sparse geometric supervision.

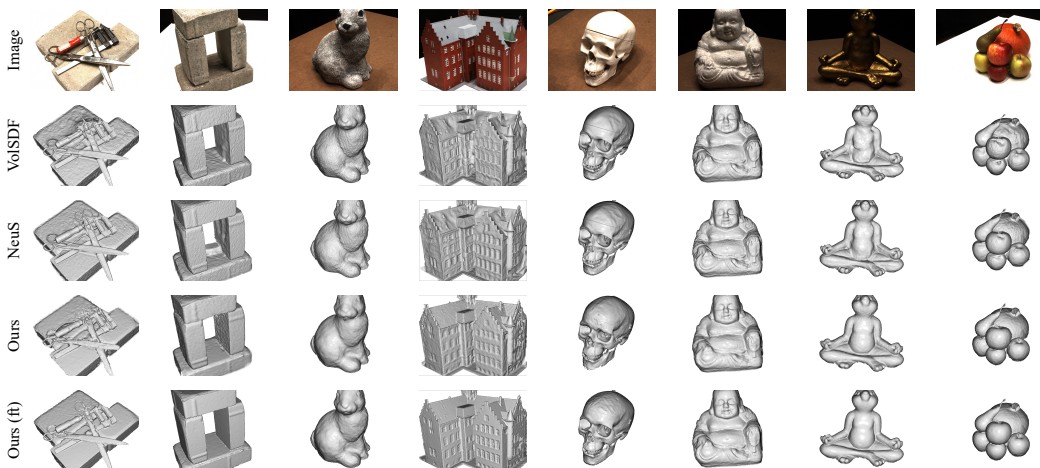

Figure 7: **Qualitative comparisons with per-scene overfitting methods on DTU dataset with dense inputs.**

**Generalizing to BlendedMVS.** Like [23], we conduct additional verification on BlendedMVS to showcase the generalization ability of our model. We also employ the same evaluation strategy as [23] for a fair comparison. As shown in Fig. 1, just through a fast network inference, our model can recover more geometric details than SparseNeuS [23]. Through the fine-tuning of 5k iterations, the effect will be significantly improved and better than SparseNeuS with 10k iterations.

### 4.2 Analysis

**Ablation studies.** We conduct ablation studies on DTU dataset to understand how the components of our model contribute to the overall performance. We start with our baseline model SparseNeuS [23], and gradually insert our contributions. The results in Tab. 4 show that our full model combining all components has the best mean score, and the baseline model, without any of our contributions, performs the worst. **Multi-scale Feature-metric Consistency (MFC):** Our self-enhanced MFC can continuously improve the multi-view consistency of the model, and we also elaborated the ablation results of its three main characteristics in Tab. 3. And the baseline is based on the pixel-wise feature consistency proposed in [65]. It can be seen that our strategy is more robust and efficiency. **Generalized Multi-scale Volume (GMV):**

| Patch-similarity | Multi-scale | Self-enhanced | Mean |
|:---:|:---:|:---:|:---:|
| ✗ | ✗ | ✗ | 1.86 |
| ✓ | ✗ | ✗ | 1.76 |
| ✓ | ✓ | ✗ | 1.73 |
| ✓ | ✓ | ✓ | **1.62** |

Table 3: **Some ablation studies on MFC.**

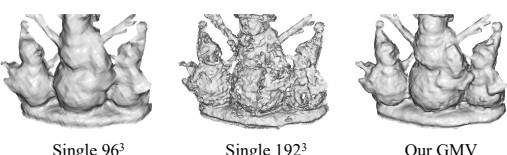

Single 96³     Single 192³     Our GMV

Figure 8: **Reconstruction from volumes with different resolution.**

| MFC | GMV | VCL | 24 | 37 | 40 | 55 | 63 | 65 | 69 | 83 | 97 | 105 | 106 | 110 | 114 | 118 | 122 | Mean |
|---|---|---|---|---|---|---|---|---|---|---|---|---|---|---|---|---|---|---|
| ✗ | ✗ | ✗ | 2.26 | 3.39 | 2.04 | 1.27 | 2.47 | 2.65 | 1.62 | 1.84 | 1.61 | 1.32 | 1.82 | 1.94 | 0.91 | 1.78 | 1.62 | 1.90 |
| ✓ | ✗ | ✗ | 1.61 | 3.12 | 1.99 | 1.16 | 2.00 | 2.21 | 1.30 | 1.58 | 1.45 | 1.18 | 1.48 | 1.53 | 0.80 | 1.54 | 1.43 | 1.62 |
| ✓ | ✓ | ✗ | 1.51 | 3.07 | 1.88 | 0.97 | 1.56 | 2.11 | 1.12 | **1.45** | **1.31** | **0.95** | 1.20 | 1.02 | 0.64 | 1.32 | 1.24 | 1.42 |
| ✓ | ✓ | ✓ | **1.45** | **2.77** | **1.69** | **0.97** | **1.54** | **1.90** | **1.03** | 1.49 | 1.36 | 0.97 | **1.07** | **0.97** | **0.62** | **1.14** | **1.16** | **1.34** |

Table 4: **Ablation results on DTU.**

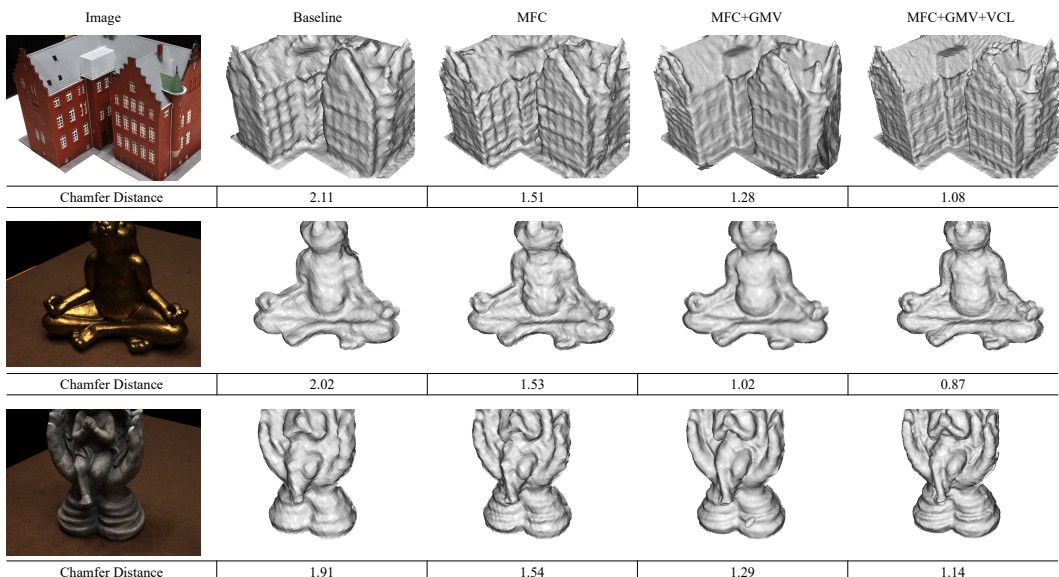

Figure 9: **Visualization of some ablation results on DTU.**

We show some results of the models have different resolutions in Fig. 8. We can see that the reconstruction of a single high-resolution volume is unbearably noisy (higher-resolution volume will lead to more empty voxels, which is more tricky for generalizable models.) and overly smooth at low resolution, whereas our GMV reconstructs clean and detailed geometry. And our representation is lighter than a single volume with a resolution of $32 \times 192^3$ due to our thin feature. **View Contrast Loss (VCL):** As the results shown in Tab. 1 and Tab. 2, the reconstruction with dense inputs is more accurate than the sparse reconstruction, we therefore treat the former as a teacher to teach the latter. The results shown in Tab. 4 validate that this strategy can indeed improve the reconstruction quality of the model. The visualization of the ablation results are shown in Fig. 9, which further depicts that every contributions we propose can continuously improve performance.

**Limitation.** Although our model exhibits excellent generalization performance in multi-view reconstruction, we found that it cannot satisfactorily handle scenes with large camera motion, such as surrounding cases. Because in these scenarios, the aggregation features will be polluted by the ray features shooting from behind. Our current solution is to first predict the local structure covered by some adjacent viewpoints like [23, 58, 36], and finally fuse them together.

## 5 Conclusion

In this paper, we introduced GenS, an end-to-end generalizable neural surface reconstruction model. We first encode all scenes into our generalized multi-scale volume, a more powerful representation that can reconstruct clean and detailed 3D structures. Then we introduce the multi-scale feature-metric consistency to combat the challenge of the photometric consistency failure. The learnable multi-scale feature can provide more discriminative representation and can be self-enhanced during the generalization training. And we finally designed a view contrast loss to improve the accuracy of the reconstruction through distilling the finer reconstruction from dense inputs to the reconstruction from sparse inputs. Experimental results on both DTU and BlendedMVS datasets show that our model possess stronger generalization ability and can achieve start-of-the-art reconstruction through fast network inference or efficient fine-tuning. In the future, we will focus on improving the performance of the model in difficult scenarios.

## Acknowledgments and Disclosure of Funding

This work is financially supported by National Natural Science Foundation of China U21B2012 and 62072013, Shenzhen Science and Technology Program-Shenzhen Cultivation of Excellent Scientific and Technological Innovation Talents project(Grant No. RCJC20200714114435057) , Shenzhen Science and Technology Program-Shenzhen Hong Kong joint funding project (Grant No. SGDX20211123144400001), this work is also financially supported for Outstanding Talents Training Fund in Shenzhen. In addition, we thank our collaborators in Alibaba Group and the anonymous reviewers for their valuable comments.

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
