# GenS: Generalizable Neural Surface Reconstruction from Multi-View Images

## (Supplemental material)

## A  Implementation details of the network

| Layer | Kernel | Channels | Out | Input |
|---|---|---|---|---|
| Conv0a | $3 \times 3$ | 3/8 | 1 | Image |
| Conv0b | $3 \times 3$ | 8/8 | 1 | Conv0a |
| Conv1a | $3 \times 3$ | 8/16 | 2 | Conv0b |
| Conv1b | $3 \times 3$ | 16/16 | 2 | Conv1a |
| Conv2a | $3 \times 3$ | 16/32 | 4 | Conv1b |
| Conv2b | $3 \times 3$ | 32/32 | 4 | Conv2a |
| Conv3a | $3 \times 3$ | 32/64 | 8 | Conv2b |
| Conv3b | $3 \times 3$ | 64/64 | 8 | Conv3a |
| Conv4a | $3 \times 3$ | 64/128 | 16 | Conv3b |
| Conv4b | $3 \times 3$ | 128/128 | 16 | Conv4a |
| **Out4** | $3 \times 3$ | 128/4 | 16 | Conv4b |
| Dconv3 | $3 \times 3$ | 128/64 | 8 | Conv4b |
| **Out3** | $3 \times 3$ | 64/4 | 8 | Dconv3 |
| Dconv2 | $3 \times 3$ | 64/32 | 4 | Dconv3 + Conv3b |
| **Out2** | $3 \times 3$ | 32/4 | 4 | Dconv2 |
| Dconv1 | $3 \times 3$ | 32/16 | 2 | Dconv2 + Conv2b |
| **Out1** | $3 \times 3$ | 16/4 | 2 | Dconv1 |
| Dconv0 | $3 \times 3$ | 16/8 | 1 | Dconv1 + Conv1b |
| **Out0** | $3 \times 3$ | 8/4 | 1 | Dconv0 + Conv0b |

Table 1: Network architecture of our image feature network.

| Layer | Kernel | Channels | Out | Input |
|---|---|---|---|---|
| Conv0a | $3 \times 3 \times 3$ | 8/8 | 2 | $B_0$ |
| Conv0b | $3 \times 3 \times 3$ | 8/8 | 2 | Conv0a |
| Conv1a | $3 \times 3 \times 3$ | 8+8/16 | 4 | $[\text{Conv0b}, B_1]$ |
| Conv1b | $3 \times 3 \times 3$ | 16/16 | 4 | Conv1a |
| Conv2a | $3 \times 3 \times 3$ | 16+8/32 | 8 | $[\text{Conv1b}, B_2]$ |
| Conv2b | $3 \times 3 \times 3$ | 32/32 | 8 | Conv2a |
| Conv3a | $3 \times 3 \times 3$ | 32+8/64 | 16 | $[\text{Conv2b}, B_3]$ |
| Conv3b | $3 \times 3 \times 3$ | 64/64 | 16 | Conv3a |
| Conv4a | $3 \times 3 \times 3$ | 64+8/128 | 32 | $[\text{Conv3b}, B_4]$ |
| Conv4b | $3 \times 3 \times 3$ | 128/128 | 32 | Conv4a |
| Dconv4 | $3 \times 3 \times 3$ | 128/64 | 16 | Conv4b |
| **Out4** | $3 \times 3 \times 3$ | 64/4 | 16 | Dconv4 |
| Dconv3 | $3 \times 3 \times 3$ | 64/32 | 8 | Dconv4 |
| **Out3** | $3 \times 3 \times 3$ | 32/4 | 8 | Dconv3 |
| Dconv2 | $3 \times 3 \times 3$ | 32/16 | 4 | Dconv3 |
| **Out2** | $3 \times 3 \times 3$ | 16/4 | 4 | Dconv2 |
| Dconv1 | $3 \times 3 \times 3$ | 16/8 | 2 | Dconv2 |
| **Out1** | $3 \times 3 \times 3$ | 8/4 | 2 | Dconv1 |
| Dconv0 | $3 \times 3 \times 3$ | 8/8 | 1 | Dconv1 |
| **Out0** | $3 \times 3 \times 3$ | 8/4 | 1 | Dconv0 |

Table 2: Network architecture of our 3D CNNs.

**Image feature network.**  In this paper, we apply the shared FPN network to extract multi-scale features for all multi-view inputs first. The detailed network architecture is shown in Tab. 1. "Out" column refers to the downscaling factor to the input image. All convolutional layers except the output layer consist of convolution block, batch normalization [3] and ReLU activator [4]. As presented in our main paper, we apply a feature network of 5 scales, and the outputs with incremental resolution of "Out4", "Out3", "Out2", "Out1" and "Out0" are the multi-scale features we need. We equipe each feature map with only 4 channels for efficiency.

**3D CNNs.**  We first construct the multi-scale cost volume from multi-scale features, and then employ a 3D CNNs to refine them to get our final multi-scale volume. As shown in Tab. 2, we inject cost volumes of corresponding scales at different stages of the encoder of the model to avoid repeatedly running the network for each scale, which greatly improves the efficiency. Similarly, we only generate volumes of 4 channels for efficiency.

**MLPs.**  We adopt the similar network architecture and initialization scheme as [9, 6]. Meanwhile, we set the number of channels in each layer of the network to 128, which is half of the original.

## B  Annealing surface sampling strategy

To further speed up the fine-tuning, we propose an annealing surface sampling strategy to reduce the number of sampling points. In the early stage, we use the same importance sampling strategy as NeuS [6] to train a decent model, which can roughly predict the surface position. Afterwards, we only sample a small number of points near the surface. Specifically, we first sample a certain number

37th Conference on Neural Information Processing Systems (NeurIPS 2023).

of points uniformly on the ray and find the surface position via Eq. 7. Then we define a sampling range with this surface position as the median, and only sample a small number of points from it for subsequent volume rendering. To avoid that the sampling range does not include the ground-truth surface due to the limited capacity of the early model, we define an annealing sampling range, which means first defining a wider sampling range and then gradually narrowing it down.

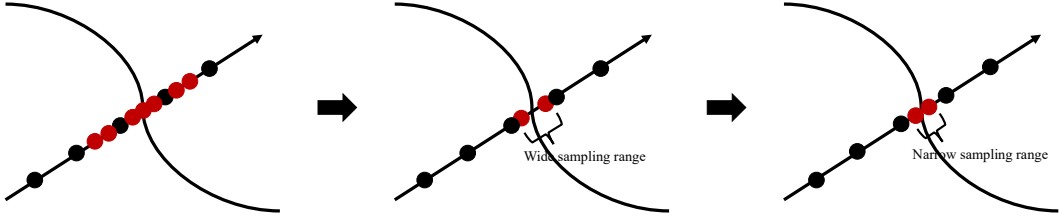

Figure 1: **Annealing surface sampling.**

## C   More ablation studies in dense seeting

Here, we show more ablation studies in dense setting.

### C.1   Generalized multi-scale volume

As analyzed in our main paper, our generalization model to reconstruct surfaces from multi-view images faces many tricky problems. The increase of viewpoints leads to more serious occlusion, and the aggregated features are easily polluted by features of occluded rays shot from behind, which leads to poor performance of the general single volume representation. In this paper, we build the generalized multi-scale volume (GMV) to efficiently represent the scene, which exploits multi-scale information in both 2D (FPN feature) and 3D spaces. We show more visual comparisons of our generalized multi-scale volume and the single large-width volume in Fig. 2. It can be seen that the surface reconstructed by the single volume method has obvious noise and holes, and our GMV can recover smoother and more refined structures. These significant advantages just demonstrate the effectiveness of our GMV.

| GMV | MFS | VCL | Mean |
|---|---|---|---|
| ✗ | ✗ | ✗ | 1.92 |
| ✓ | ✗ | ✗ | 1.08 |
| ✓ | ✓ | ✗ | 0.83 |
| ✓ | ✓ | ✓ | **0.81** |

Table 3: **Ablation on the number of scales.**

| Number of scales | 24 | 37 | 105 |
|---|---|---|---|
| 3 | 0.87 | 1.59 | 0.80 |
| 5 | 0.66 | 1.01 | 0.78 |

Table 4: **Ablation on the number of scales.**

The volume with small resolution has more global information, which is conducive to maintaining global smoothness, while the large-resolution volume restores geometric details through finer features. We further conduct an ablation study on the number of scales. We remove 2 low-resolution volumes to form the 3-scale GMV in Tab. 4, whose resolution is increased from $2^6$ to $2^8$. The results show that the 5-scale GMV has a great advantage over the 3-scale GMV, which verifies the effectiveness of the multi-scale design.

### C.2   Multi-scale feature-metric consistency

Our multi-scale feature-metric consistency (MFC) is conducted on the multi-scale feature space, which can provide a more discriminative representation and enable more robust matching compared with the ordinary photometric consistency. To demonstrate our superiority, we perform some experiments to compare with photometric consistency as mentioned in our main paper. We have shown some visual comparisons in Fig. 9, and we illustrate more qualitative and quantitative results here. The "Base" in Tab. 5 is a model with only the generalized multi-scale volume. The "PC" stands for the model applying

| Method | 24 | 37 | 105 |
|---|---|---|---|
| Base | 0.92 | 1.91 | 1.09 |
| +PC | 2.10 | 2.39 | 1.24 |
| +MPC | 0.82 | 1.74 | 0.85 |
| +MFC | 0.68 | 1.12 | 0.77 |

Table 5: **Ablation on the multi-scale feature-metric consistency.**

the photometric consistency, which constrain the multi-view consistency in the image space as [1, 2]. The results show that it cannot work well for generalization training. We analyze that the main reason

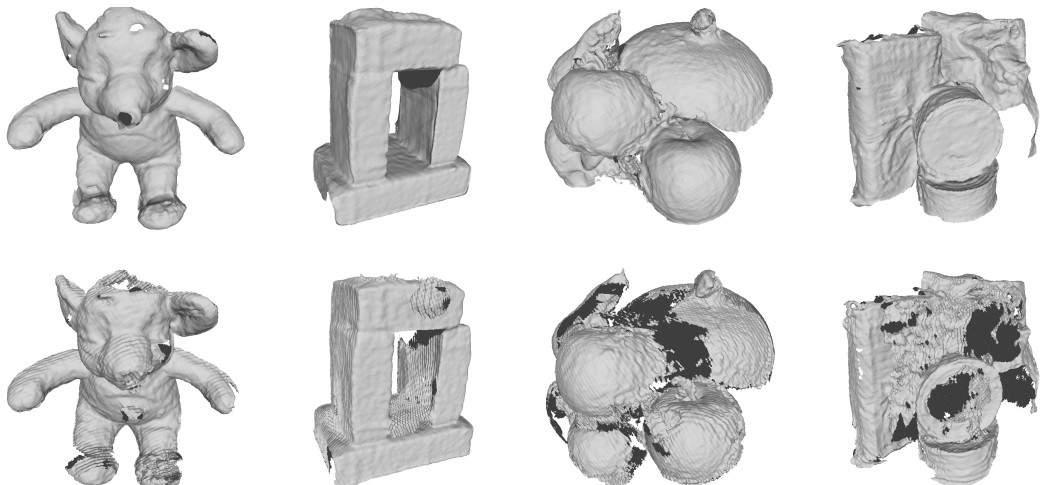

Figure 2: **Visual comparison of our GMV and the single large-width volume.** The first row is the result of our GMV, and the second row is the result of the single volume.

is that the region where the photometric consistency fails is more challenging for generalization training. Some SOTA self-supervised multi-view stereo methods (Self-MVS) [7, 8, 5] try to mask out those regions to avoid polluting model training. Another advantage they have over our model is that they enforce photometric consistency at the image level, which can impose global geometric smoothness constraints, whereas our model can only handle local image patches individually due to its reliance on ray sampling. Based on this intuition, we attempt to increase the receptive field of image patches, that is, we downsample the image in the early stage, and then sample the image patch for multi-view matching. As the model converges, we gradually upsample the image to the original resolution. We call this strategy multi-scale photometric consistency (MPC). The results in Tab. 5 show that enlarging the receptive field works well for our generalization training and brings improvements to the base model. In addition to the multi-scale information, we find that learnable features can provide more discriminative representations, which is of great benefit for improving matching accuracy. Therefore, we utilize the off-the-shelf multi-scale features generated from our FPN feature network to achieve our multi-scale feature-metric consistency, which simultaneously have different ranges of receptive fields. The results shown in Tab. 5 just prove the effectiveness of our MFC. We show more visual comparison results in Fig. 4 to further verify the advantages of our MFC. It can be seen that our MFC can reconstruct finer and smoother structures, especially in low-texture and complex scenes.

## C.3    View contrast loss

We have shown in our main paper that the view contrast loss can improve the reconstruction of student models. As shown in Tab. 3, it can also benefit to the reconstruction of the teacher model. Some visual comparison in Fig. 3 shows that the view contrast loss can improve the smoothness of reconstruction especially for those regions visible by few viewpoints.

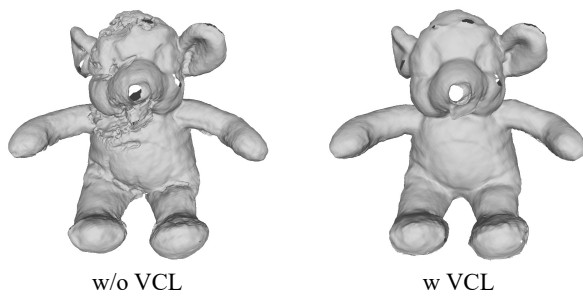

w/o VCL                    w VCL

Figure 3: **Visual comparison of the model with VCL and without VCL.**

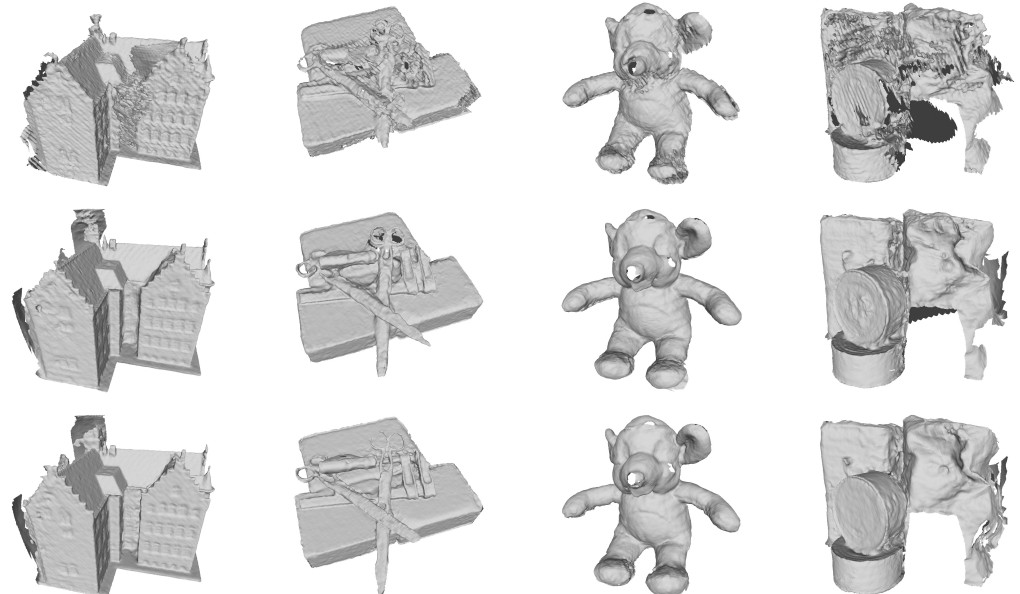

Figure 4: **Visual comparison of our MFC, MPC and PC.** The first row is the result of PC, the second row is the result of MPC, and the last row is the results of MFC.