# OpenReview forum: "GenS: Generalizable Neural Surface Reconstruction from Multi-View Images"
_NeurIPS.cc/2023/Conference — NeurIPS 2023 poster_

### Official Review · Reviewer_XxbL · 2023-06-30

**Soundness:** 3 good
**Presentation:** 3 good
**Contribution:** 3 good
**Rating:** 6
**Confidence:** 5

**Summary:**

This paper proposed a method for neural implicit surface reconstruction from multi-view images. The proposed method is generalizable without per-scene overfitting. The key component of the proposed method is a multi-view feature volume constructed from a multi-scale FPN network. During training process, the multi-scale feature is used to constrain the view-consistency as well as geometry smoothness. Experiment results show that the proposed method achieves better results than some previous pre-scene methods and achieve state-of-the-art results on DTU dataset with finetuning.

**Strengths:**

- A new approach that targets generalizable neural surface representation is interesting. The proposed method is reasonable and technical sound.
- Writing is overall clear.
- Result quality is not bad.

**Weaknesses:**

- The current proposed method (from my POV) is a direct extension of SparseNeuS with more input view and multi-scale volumes (and two additional loss term that using multi-scale features). Neural geometry representation with multi-scale approaches has been also proposed in [1] (BTW this was also missing in related works of this paper). The combination of these methods has not appeared before, but the overall novelty would still be limited, as the paper does not demonstrate the key challenges (either theoretical or practical) when combining these techniques. It more sounds like "we put this and that together, and it works better than previous methods."
- Experiment is a little bit weak. Quantitative results are only reported on DTU dataset. From results on Tab.1, retraining Geo-NeuS with the same dense-view input works really well (almost same as GeNIS with finetune). Fig.8 demonstrates qualitative results on BlendedMVS dataset; however, it is hard to say which one is better for Ours(ft) and SparseNeuS(ft). These results suggest the improvement of the proposed method (especially, over its most related baseline, SparseNeuS) is marginal.

[1] Takikawa, Towaki, et al. "Neural geometric level of detail: Real-time rendering with implicit 3D shapes." Proceedings of the IEEE/CVF Conference on Computer Vision and Pattern Recognition. 2021.


**Questions:**

- How long does it take for finetuning? A long finetuning process would make the proposed method less interesting for "generalizable" representation.
- I still failed to understand why the proposed view-contrast loss can improve smoothness. Is the teacher volume and student volume constructed with the same network? And why add consistency between student and teacher can improve smoothness?
- I notice that ablation studies are only conducted on three data points instead of full set in DTU dataset. Is there any reason?
- Is multi-view volume a must for generalizable representations? Current experiments only demonstrate multi-view feature consistency is useful. What about using single volume but with all the feature consistency and contrast constraints? i.e., a naive extension of SparseNeuS with the loss proposed in the paper.

**Limitations:**

The limitations and societal impact have been discussed in the paper.

---

> ### Author Rebuttal · Authors · 2023-08-08
>
> We sincerely thank the reviewer XxbL for your time and valuable comments. Here, we will explain your concerns point by point. Due to words limitation, we put most experiments in the uploaded PDF.
>
> **Q1: Missing the citation of [1], which also adopts a multi-scale strategy**
>
> Sorry for missing this important citation since we only focused on the volume rendering-based methods, and we will cite this paper in our final version. Meanwhile, our method and [1] are quite different: our multi-scale volume are constructed from the image feature, and that of [1] are learnable parameters.  We will cite [1] in our final version.
>
> [1] Neural geometric level of detail: Real-time rendering with implicit 3D shapes, CVPR 2021.
>
> **Q2: There is a lack of the demonstration of the key challenges when designing modules.**
>
> We sincerely thank the reviewer for this suggestion, which we believe is very helpful in enhancing the persuasiveness of our article. In fact, due to the limitation of the page and the rush of deadline, we only show the main implementation and main experiments in the paper, and put many necessity analyzes of each modules in the Supp. We will adjust the structure of the article and enrich the analysis of key challenges in the final version according to your suggestions. We mainly focus and deal with three key challenges: 1) how to make the model recover geometric details while maintaining global shape consistency. 2) how to reconstruct the geometry that is consistent across different views. 3) how to make the model be robust to those regions only visible to few views. We have positively proposed generalized multi-scale volume, multi-scale feature-metric consistency and view contrast loss to solve these problems respectively, and verified their effectiveness through extensive experiments.
>
> **Q3: Experiment is a little bit weak. Quantitative results are only reported on DTU dataset.**
>
> Most of the previous methods including our baseline NeuS and SparseNeuS only conduct the quantitative results on DTU dataset, and show the qualitative results on BlendedMVS dataset. We just follow the same evaluation strategy as them.
>
> **Q4: From results on Tab.1, retraining Geo-NeuS with the same dense-view input works really well (almost same as GeNIS with finetune).**
>
> You overlooked one important factor of the training time, which is one advantage of our model. Our model only needs to be fine-tuned in 5K iterations per-scene (about 20 mins), while Geo-NeuS needs to be trained in 300K iterations (about 10 hours).
>
> **Q5: Fig.8 demonstrates qualitative results on BlendedMVS dataset; however, it is hard to say which one is better for Ours(ft) and SparseNeuS(ft). These results suggest the improvement of the proposed method (especially, over its most related baseline, SparseNeuS) is marginal.**
>
> 1. Actually, in the results depicted in Fig. 8, Our qualitative results are significantly superior to SparseNeuS with or without fine-tuning. In order to see them more clearly, we purposely zoomed in on the local details as shown in Figure 1 in the PDF we uploaded. From which we can clearly see that our reconstruction results are superior, especially in details such as eyes, ears and nose.
> 2. To further demonstrate that the improvement of our model is not marginal, we conduct the quantitative comparison with existing methods SparseNeuS using both dense and sparse inputs. As the results shown in the table in our global response channel and Table 2 in our uploaded PDF, our model can achieve more than 60% improvement than SparseNeuS with dense inputs, and more than 20% with sparse inputs.
>
> **Q6: How long does it take for fine-tuning?**
>
> As described in our paper, we fine-tune each scene for about 20 minutes. And even with only about 10 minutes of fine-tuning, we can already achieve very desirable performance and outperform most SOTA methods.
>
> **Q7: Why the proposed view-contrast loss can improve smoothness?  Is the teacher volume and student volume constructed with the same network?**
>
> 1. As we state in our paper, there are regions visible only to a few viewpoints, which are easily polluted by irrelevant rays when aggregating features from multiple viewpoints, leading to geometric prediction errors. We show an naïve example in Figure 2 in our uploaded PDF. To make the model be robust to these tricky regions, we use results from dense views to teach results from the randomly selected sparse views. In this way, our model can infer more accurate geometry on these regions and make the reconstruction smoother. Especially when the resolution of the mesh is high, this smoothing effect is more obvious, as shown in Figure 2 in our uploaded PDF.
> 2. Yes, we use the shared network for student volume and teacher volume.
>
> **Q8: Why only conduct ablation studies on three scenes?**
>
> The reason is because of page limitations and layout considerations. We randomly select the results of three scenes, which can already demonstrate the effectiveness of each module. We show you the results of all scenes in Table 1 in our uploaded PDF.
>
> **Q9: Is multi-scale volume a must for generalizable representations? Current experiments only demonstrate multi-view feature consistency is useful. What about using single volume but with all the feature consistency and contrast constraints? i.e., a naive extension of SparseNeuS with the loss proposed in the paper.**
>
> 1. The generalized multi-scale volume (GMV) indeed plays an important role in generalization training, and we have proved this through the ablation analyzes shown in our main paper and Supp, e.g., our GMV can be improved by more than 40% from 1.92 to 1.08 compared with single high-scale volume model as shown in  Table 1 in our uploaded PDF.
> 2. As your suggestion, we further conduct experiments combining our proposed loss with the single-volume model SparseNeuS. The results are shown in Table 3 in our uploaded PDF.  It can bee seen that our innovation can indeed lead to significant improvements.

---

> > ### Comment · Reviewer_XxbL · 2023-08-12
> > **Reply to rebuttal**
> >
> > Thanks for the detailed rebuttal, here is my comments after reading the rebuttal.
> > - Related work (Q1). I appreciate the rebuttal discussed the differences between this work and the mentioned work. I would like to additionally emphasize that the Neural LOD contains octree structures that are not easy to construct from the image feature (which seems to be an advantage of this proposed method).
> >
> > - Key challenges and method significance (Q2, Q5, Q6, Q9): I am now convinced that the proposed multi-scale view-contrastive loss is useful and interesting. It seems this strategy helps the learnt representations robust to sparse inputs.
> >
> > - Experiments (Q3, Q4, Q7, Q8). I appreciate clarifications in the rebuttal.
> >
> > Overall, I feel positive after reading the rebuttal and I am willing to increase my score to "Weak Accept". Thanks.

---

> > > ### Author Response · Authors · 2023-08-14
> > >
> > > We sincerely appreciate your recognition of our work and we will revise and enrich our paper according to your constructive suggestions.

---

### Official Review · Reviewer_cro4 · 2023-07-01

**Soundness:** 4 excellent
**Presentation:** 2 fair
**Contribution:** 3 good
**Rating:** 7
**Confidence:** 4

**Summary:**

This paper proposed a generalizable Nerf method. The novelty of this method lies in a proposed multi-scale feature-metric consistency loss and a proposed view contrast loss. The feature-metric consistency loss encourages the features extracted from different images at the same surface point to be consistent with each other. Doing this in feature space is claimed to be more discriminative. The view contrastive loss enforces that the recovered surfaces from subsampled image sets be aligned with that of a full image set, to encourage smoothness n surface regions with poor visibility. This method can be applied directly to predict surfaces or have its output fine-tuned for a short time to arrive at a stronger final solution. With this method, the authors achieve state-of-the-art results on the DTU and BlendedMVS datasets.

**Strengths:**

The strengths of this paper are that the proposed losses are novel and well-founded and that their inclusion in a generalizable Nerf pipeline leads to state-of-the-art results both qualitatively and quantitatively.

**Weaknesses:**

The main weakness of the paper is the grammar and sentence structure. I would heavily advise that some polishing be done to the paper before the final submission. Other than that the proposed method seems sound and the results are impressive.



**Questions:**

Some requests I have for the paper:
-> Provide a numerical ablation over removing the proposed losses
-> Provide standard deviations in the means for Table 1
-> Bold the best-performing model in each column for Table 1

With respect to comparing to prior works, why do you not provide the performance of models such as SparseNeus and PixelNerf in Table 1? You make claims that these will perform worse, but this is not demonstrated quantitatively. Please provide the results of training at least one other generalizable method with the same fine-tuning process applied to your method in this table.

**Limitations:**

-

---

> ### Author Rebuttal · Authors · 2023-08-08
>
> We sincerely thank the reviewer cro4 for recognizing our work and the valuable comments. Here, we will explain your concerns point by point.
>
> **Q1: The grammar and sentence structure should be further polished.**
>
> Thanks for pointing them out, and we appreciate these suggestions. We will seriously enrich and correct our paper in the final version.
>
> **Q2: Provide a numerical ablation over removing the proposed losses.**
>
> Due to the rush of the deadline and the limitation of the page, we only conduct the ablation studies of our main innovations, e.g., view contrast loss and multi-scale feature-metric consistency loss. We include more ablation analysis on the Supp., but some ablation experiments on regularization loss, i.e., sparse loss $L_{sp}$, smooth loss $L_{sm}$ and total variation loss $L_{tv}$ are still ignored as you point out. Here, we further append these ablation experiments. To quickly complete these experiments within a limited rebuttal time, we perform the experiments on the model with single small feature volume and 3 views as input, which is just like the first stage of SparseNeuS. The results are shown in the table below, we can see that each regularization term can steadily improve the model performance. And the ablation for other losses are shown in Tab.2 of our main paper.
>
> |  | 24 | 37 |  40 | 55 | 63 | 65 | 69 | 83 | 97 | 105 | 106 | 110 | 114 | 118 | 122 | Mean |
> | --- | --- | --- | --- | --- | --- | --- | --- | --- | --- | --- | --- | --- | --- | --- | --- | --- |
> | Base | 2.17 | 3.50 | 2.45 | 1.36 | 2.64 | 2.51 | 1.75 | 1.92 | 1.79 | 1.43 | 1.89 | 2.03 | 1.05 | 2.01 | 1.71 | 2.01 |
> | +$L_{sp}$ | 2.12 | 3.50 | 2.15 | 1.34 | 2.52 | 2.67 | 1.72 | 1.95 | 1.70 | 1.39 | 2.05 | 2.01 | 1.03 | 1.84 | 1.66 | 1.98 |
> | +$L_{sp}$+$L_{sm}$ | 2.11 | 3.43 | 2.14 | 1.32 | 2.50 | 2.59 | 1.69 | 1.95 | 1.72 | 1.38 | 1.88 | 2.02 | 0.97 | 1.91 | 1.60 | 1.95 |
> | +$L_{sp}$+$L_{sm}$+$L_{tv}$ | 2.26 | 3.39 | 2.04 | 1.27 | 2.47 | 2.65 | 1.62 | 1.84 | 1.61 | 1.32 | 1.82 | 1.94 | 0.91 | 1.78 | 1.62 | 1.90 |
>
> **Q3: Provide standard deviations in the means for Tab. 1.**
>
> Thanks for this constructive suggestion, it’s an important strategy to support our model. Since it takes a long time to retrain multiple sets of generalizable models, in this limited rebuttal time, we do multiple sets of fine-tuning experiments  to obtain the variance of the fine-tuning results, and the variance of the generalizable model will be supplemented in the final version later. We fine-tune each scene four more times, and the results are shown in the Table below. It can be seen that our model can get an performance of about 0.62 after fine-tuning, and the ones with relatively large standard deviations are mostly concentrated in difficult scenes.
>
> |  | 24 | 37 |  40 | 55 | 63 | 65 | 69 | 83 | 97 | 105 | 106 | 110 | 114 | 118 | 122 | Mean |
> | --- | --- | --- | --- | --- | --- | --- | --- | --- | --- | --- | --- | --- | --- | --- | --- | --- |
> | Exam1 | 0.55 | 0.71 | 0.39 | 0.38 | 0.79 | 0.65 | 0.57 | 1.29 | 0.96 | 0.64 | 0.49 | 0.59 | 0.33 | 0.44 | 0.45 | 0.615 |
> | Exam2 | 0.54 | 0.74 | 0.40 | 0.38 | 0.82 | 0.62 | 0.61 | 1.35 | 0.94 | 0.64 | 0.50 | 0.64 | 0.32 | 0.44 | 0.43 | 0.625 |
> | Exam3 | 0.57 | 0.71 | 0.40 | 0.39 | 0.77 | 0.63 | 0.58 | 1.32 | 0.94 | 0.65 | 0.49 | 0.60 | 0.34 | 0.45 | 0.45 | 0.619 |
> | Exam4 | 0.56 | 0.73 | 0.41 | 0.37 | 0.79 | 0.69 | 0.56 | 1.30 | 0.97 | 0.67 | 0.52 | 0.62 | 0.33 | 0.43 | 0.44 | 0.626 |
> | Exam5 | 0.55 | 0.72 | 0.38 | 0.37 | 0.80 | 0.64 | 0.59 | 1.29 | 0.94 | 0.64 | 0.51 | 0.61 | 0.34 | 0.44 | 0.44 | 0.617 |
> | Mean | 0.554 | 0.722 | 0.396 | 0.378 | 0.794 | 0.646 | 0.582 | 1.310 | 0.950 | 0.648 | 0.502 | 0.612 | 0.332 | 0.440 | 0.442 | 0.621 |
> |Std | 0.011 | 0.013 | 0.011 | 0.008 | 0.018 | 0.027 | 0.019 | 0.025 | 0.014 | 0.013 | 0.013 | 0.019 | 0.008 | 0.007 | 0.008 | 0.004 |
>
> **Q4: Bold the best-performance model in each column for Table 1.**
>
> Thanks for this valuable suggestion, and we will revise our paper according to your suggestions.
>
> **Q5: There are no quantitative comparison with existing generalizable methods like SparseNeuS, and please provide these results.**
>
> Sorry for missing these important quantitative comparisons, and we only do qualitative comparisons on BlendedMVS dataset. Here, we conduct two groups of experiments in the case of sparse and dense input to fully prove our advantages.
>
> 1. Sparse input: We adopt the same sparse configuration as SparseNeuS and only train our model for10W iterations, which is half of SparseNeuS, to save rebuttal time. The results are shown in Table 2 in our uploaded PDF. We can see that our model has a significant advantage over existing methods, e.g., our fine-tuned model can outperform SparseNeuS by more than 20%..
>
> 2. Dense input: We train our model with dense inputs to reconstruct complete geometry. We implemented SparseNeuS in two configurations, one with feature volume resolution of 96 and one with resolution of 192. And we train all models with the same procedure. The results are shown in the table below. We can see that our model can outperform SparseNeuS by more than 60%.
> |  | 24 | 37 |  40 | 55 | 63 | 65 | 69 | 83 | 97 | 105 | 106 | 110 | 114 | 118 | 122 | Mean |
> | --- | --- | --- | --- | --- | --- | --- | --- | --- | --- | --- | --- | --- | --- | --- | --- | --- |
> | SparseNeuS (96 res) |2.41|3.11| 2.14|1.22|2.38|2.80|1.70|2.64| 2.15 | 1.68 | 1.52 | 2.80 | 1.04 | 1.91 | 1.72 | 2.08 |
> | SparseNeuS (192 res)|3.70|2.87|1.91|1.03|1.72|1.67|1.59|2.35|2.03|1.40|1.33|3.57|0.72|1.37|1.52|1.92|
> | Our GeNIS | 0.66|1.01|0.71|0.43|1.06|0.99|0.73|1.43|1.18|0.78|0.64|0.93|0.38|0.54|0.54|0.80|

---

> > ### Comment · Reviewer_cro4 · 2023-08-10
> > **reply**
> >
> > My comments and questions have been adequately addressed. Thank you for taking the time to do so.  I believe my score is still fair for the paper.

---

> > > ### Author Response · Authors · 2023-08-11
> > >
> > > We sincerely thank the reviewer cro4 for recognizing our work and we will revise and enrich our paper according to your constructive suggestions.

---

### Official Review · Reviewer_EdqW · 2023-07-06

**Soundness:** 3 good
**Presentation:** 3 good
**Contribution:** 2 fair
**Rating:** 6
**Confidence:** 3

**Summary:**

This paper aims to address the problem of multi-view reconstruction using neural implicit representation. The key to improving both efficiency and efficacy is a deep network that predicts a generalizable multi-scale feature volume encapsulating the geometry and appearance of the scene given a set of posed images. The generalizable feature volume circumvent the need of optimizing the neural imiplicit representation from the scratch for each testing scene. The authors also devise two losses -- multi-view feature consistency and view contrast loss -- to further improve the reconstruction performance. Experiments demonstrate that the proposed method outperform several state-of-the-art methods in the DTU dataset.

**Strengths:**

- View contrast loss is novel and interesting in the context of NeRF training.

- Ablation studies on multi-scale feature volume, , multi-view feature consistency, and view contrast loss validates the contribution of these critical components in the proposed system.

**Weaknesses:**

- My major concern of this paper is that the key ideas presented have been explored separately in several works.
    1. Multi-scale feature volume for neural implicit field has been widely used in MonoSDF, NGP, and several other works. Generalizable multi-scale feature volume has been a go-to representation for deep learning based Multi-view Stereo.
    2. Feature consistency has been investigated in MVSDF.
    3. The surface approximation method is borrowed from Geo-NeuS.
As a result, the novelties are somewhat limited.

- What is the annealing factor in (eq. 5) in training? I understand the authors provided a detailed description of the annealing factor for fine-tuning on a particular scene, but I wonder how to control the annealing factor in a batch-wise training setup.

- Some questions regarding experiments:
    1. The baseline results in Table 2 are surprisingly low. Without equipping all the novelties presented in the paper, the baseline seems to be a NeuS with a large feature volume. Can the authors explain why this baseline is much worse than NeuS (e.g., 3.7 vs 1 in scene 24 in DTU)?
    2. Missing comparisons to recent efficient neural implicit methods. Although works like Voxurf do not have a generalizable deep network to obtain learnt feature volume given a new scene, they also claim to achieve state-of-the-art results on DTU after optimizing on a scene for several minutes.


**Questions:**

See questions in the weaknesses section.

**Limitations:**

Yes

---

> ### Author Rebuttal · Authors · 2023-08-08
>
> We sincerely thank the reviewer EdqW for recognizing our work and the valuable comments. Here, we will explain your concerns point by point.
>
> **Q1: The multi-scale mechanism has been explored in existing method.**
>
> The multi-scale strategy is effective and has inspired and advanced many fields. We are the first to apply the multi-scale strategy to generalizable neural implicit surface reconstruction to extend SparseNeuS, and the similar strategy has also been proposed in the recent generalizable work ReTR [1]. Although the recent MonoSDF and NGP also use multi-scale ideas, they differ from our method in the following ways: 1) Our feature volume is constructed from image features through a model inference, which has strong generalization, while Both MonoSDF and NGP require scene-by-scene learning. 2) Our multi-scale feature volume can be sparsed with the learned generalization prior to further reduce storage and improve fine-tuning speed.
>
> [1] Rethinking Rendering in Generalizable Neural Surface Reconstruction: A Learning-based Solution, arxiv 2023.
>
> **Q2: The difference between our self-enhanced multi-scale feature consistency and the feature consistency in MVSDF**
>
> The feature consistency of MVSDF and ours are quite different. Our self-enhanced multi-scale feature-metric consistency (MFC) has three main advantages.
> 1) Our feature-metric consistency can be self-enhanced during the training. The robust feature consistency can train a better reconstruction model, and the better reconstruction model can extract better feature spaces, leading to more robust feature-metric consistency. To demonstrate this, we add a experiment using feature consistency without self-enhancement (pre-trained features). The results in the table below indicates that our consistency has significant advantage.
> 2) Our consistency is based on the local planar assumption and is measured by the structural similarity losses within patches like SSIM or NCC, which is more robust and effective than the pixel-wise L1 loss used in MVSDF. As the comparison shown in the table below, the model with pixel-wise L1 feature loss like MVSDF performs worse than ours.
> 3) Our consistency is measured in a multi-scale manner. The high-resolution feature space can provide finer and context-complete information, while the low-resolution feature space has larger receptive field to achieve more discriminative matching as shown in our Fig. 4. The comparison of single scale models with ours again demonstrates the superiority of our method.
>
> Note that the structure of the baseline is just like SparseNeuS, which has a single 96-resolution volume, and all experiments are conducted with three images as input.
>
> ||24|37|40|55|63|65|69|83|97|105|106|110|114|118|122|Mean|
> | --- | --- | --- | --- | --- | --- | --- | --- | --- | --- | --- | --- | --- | --- | --- | --- | --- |
> | Baseline |2.26|3.39|2.04|1.27|2.47|2.65|1.62|1.84|1.61|1.32|1.82|1.94|0.91|1.78|1.62|1.90|
> | Pixel-wise L1 |2.11|3.29|2.20|1.26|2.39|2.54|1.62|1.78|1.59|1.29|1.76|2.01|0.88|1.66|1.58|1.86|
> | Single-scale |1.64|3.39|2.16|1.27|2.29|2.25|1.48|1.76|1.55|1.17| 1.70 | 1.75 | 0.86 | 1.65 | 1.54 | 1.76 |
> | No self-enhanced | 1.77 | 3.12 | 2.03 | 1.21 | 2.09 | 2.34 | 1.49 | 1.74 | 1.54 | 1.26 | 1.60 | 1.76 | 0.91 | 1.61 | 1.48 | 1.73 |
> | Our MFC | 1.51 | 3.22 | 1.99 | 1.16 | 2.00 | 2.21 | 1.30 | 1.58 | 1.45 | 1.18 | 1.48 | 1.53 | 0.80 | 1.54 | 1.43 | 1.62 |
>
> **Q3: The surface approximation method is borrowed from Geo-NeuS.**
>
> In our multi-scale feature-metric consistency, we do not focus on the surface approximation method, but on applying the multi-scale feature space to achieve more robust matching among multi-view images. Actually, our solution is general and can be applied to any surface approximation method.
>
> **Q4: How to control the annealing factor in a batch-wise training setup?**
>
> We adopt the same strategy as NeuS and SparseNeuS , setting the annealing factor (s) as a learnable parameter, and as the network converges, (1/s) will approach 0.
>
> **Q5: Why the baseline (NeuS with a large feature volume) is much worse than NeuS?**
>
> 1. Table 2 shows the generalization results of the model trained on the training set, and it is reasonable to be worse than NeuS, an overfitting model that has been optimized for a long time in each scene.
> 2. We did find that a large volume does not work well in generalization training. The main reason we think is that the receptive field of each voxel in a large feature volume is small, and the number of the empty voxel in a scene is large, which is even worse in the large volume. This makes it difficult to train the feature extraction network. The results of the small volume are also not satisfactory due to the lack of high frequency detail. Our experiments in the able below just verfy these, and it can be seen that our multi-scale volume is the best. Note that these experiments are based on dense inputs.
>
> ||24|37|40|55|63|65|69|83|97|105|106|110|114|118|122|Mean|
> | --- | --- | --- | --- | --- | --- | --- | --- | --- | --- | --- | --- | --- | --- | --- | --- | --- |
> | SparseNeuS (96 res)|2.41|3.11|2.14|1.22|2.38|2.80|1.70|2.64|2.15|1.68|1.52|2.80|1.04|1.91|1.72|2.08|
> | SparseNeuS (192 res)|3.70|2.87|1.91|1.03|1.72|1.67|1.59|2.35|2.03|1.40|1.33|3.57|0.72|1.37|1.52|1.92|
> | Our GMV |0.92|1.91|1.12|0.77|1.37|1.21|1.09|1.55|1.35|1.09|0.96|1.14|0.41|0.70|0.67|1.08|
>
> **Q6: Missing comparisons to recent methods like Voxsurf.**
>
> Sorry for missing this comparison due to the fact that Voxsurf is not a generalizable model. Voxsurf is a very valuable work, which can greatly speed up the training compared to previous methods. Even though, our model has two main advantages than Voxsurf: 1) Our model is generalizable which can infer the geometry  just through a fast network inference and achieve satisfactory results. 2) In a specific scene, we can achieve much better performance after fine-tuning for similar periods of time. We will cite Voxsurf in our final version.

---

> > ### Comment · Reviewer_EdqW · 2023-08-14
> >
> > My concerns have been addressed adequately in the rebuttal. I will update my score shortly.

---

> > > ### Author Response · Authors · 2023-08-16
> > >
> > > We sincerely thank your recognition and we will revise and enrich our paper according to your constructive suggestions.

---

### Official Review · Reviewer_Tu79 · 2023-07-07

**Soundness:** 3 good
**Presentation:** 3 good
**Contribution:** 3 good
**Rating:** 5
**Confidence:** 4

**Summary:**

This paper proposes a generalisable neural surface reconstruction approach that is able to learn an SDF function from volume features and does not need per-scene training.

**Strengths:**

1. This paper uses a multi-scale volume to model the feature field.
2. The view contrast loss is novel to me which helps to enhance the reconstruction quality over regions that lack multi-view observations.

**Weaknesses:**

1. This paper does not mention some of the latest advanced approaches that also work on the generalisable neural surface reconstruction:
     VolRecon: Volume Rendering of Signed Ray Distance Functions for Generalizable Multi-View Reconstruction
     Rethinking Rendering in Generalizable Neural Surface Reconstruction: A Learning-based Solution
2. The experiments are not complete. It does not compare with the two SOTA methods mentioned above. Also it does not compare with the learned MVS methods, such as MVSNet.
3. The results of the reconstruction without fine-tuning is actually not looking good enough to me. For example, in figure 8, the mesh results are quite noisy.

**Questions:**

1. In Eq. 2, the MLP also takes position as input. Doesn't it make the function not generalisable to other scenes, since position is scene-specific?
2. In Eq. 3, does it make sense to blend the colors of all view projections? What if the point is not visible in certain views?
3. Eq. 10 is proposed to encourage multi-view feature consistency. But how will it handle occlusion cases? If there are occlusions, the features are naturally inconsistent.

**Limitations:**

Limitations are already revealed in the paper.

---

> ### Author Rebuttal · Authors · 2023-08-08
>
> We sincerely thank the reviewer Tu79 for recognizing our work and the valuable comments. Here, we will explain your concerns point by point.
>
> **Q1: Lacking comparison with latest advanced approaches: VolRecon [1] and ReTR [2].**
>
> Thanks for your suggestion, and the reasons why we don’t compare with these two methods are:
> 1) these two methods are contemporaneous with ours, and ReTR was made public after the NeurIPS deadline.
> 2) they are not unsupervised methods like ours, which require the ground-truth depth for supervision.
> 3) they are target at the sparse adjacent views, and only reconstruct part of the geometry.
>
> Even though, we think these comparisons are valuable and we conduct these comparisons here. For fair comparisons, we adopt the same training and testing configuration with SparseNeuS . Due to the rush of the rebuttal, we only train our model for 10W iterations, which is half of SparseNeuS, and we will supplement the results with the same training iterations as SparseNeuS in our final version. The results are shown in the table below. Without any fine-tuning, our model can achieve superior performance than existing unsupervised method SparseNeuS, and perform a little worse than supervised method (VolRecon and ReTR). Through fast fine-tuning, our model can achieve the state-of-the-art performance and even outperform the supervised methods. Note that this experiment is performed under the test set split of SparseNeuS, where each scene contains only three images. And we will cite these works in our final version.
>
> |  | 24 | 37 |  40 | 55 | 63 | 65 | 69 | 83 | 97 | 105 | 106 | 110 | 114 | 118 | 122 | Mean |
> | --- | --- | --- | --- | --- | --- | --- | --- | --- | --- | --- | --- | --- | --- | --- | --- | --- |
> | VolRecon [1] | 1.20 | 2.59 | 1.56 | 1.08 | 1.43 | 1.92 | 1.11 | 1.48 | 1.42 | 1.05 | 1.19 | 1.38 | 0.74 | 1.23 | 1.27 | 1.38|
> | ReTR [2] | 1.05 | 2.31 | 1.44 | 0.98 | 1.18 | 1.52 | 0.88 | 1.35 | 1.30 | 0.87 | 1.07 | 0.77 | 0.59 | 1.05 | 1.12 | 1.17 |
> | SparseNeuS (20W)	 | 1.68 | 3.06 | 2.25 | 1.10 | 2.37 | 2.18 | 1.28 | 1.47 | 1.80 | 1.23 | 1.19 | 1.17 | 0.75 | 1.56 | 1.55 | 1.64 |
> Ours (10W) | 1.28|3.04|2.05|0.97|1.38|2.16|1.07|1.41|1.36|0.93|1.22|1.02|0.64|1.24|1.25|1.40 |
> SparseNeuS (ft) | 1.29 | 2.27 | 1.57 | 0.88 | 1.61 | 1.86 | 1.06 | 1.27 | 1.42 | 1.07 | 0.99 | 0.87 | 0.54 | 1.15 | 1.18 | 1.27 |
> Ours (ft) | 0.75 | 1.99 | 1.34 | 0.75 | 1.12 | 1.68 | 0.77 | 1.21 | 0.98 | 0.74 | 0.89 | 0.54 | 0.43 | 0.82 | 0.91 | 0.99 |
>
> **Q2: Lacking the comparison with learning-based MVS method.**
>
> Following your suggestion, we add the comparison with VisMVSNet [3], which is an efficient learning based method. The comparison is shown in the table below. It can be seen that our model can achieve superior results even without fine-tuning.
>
> |  | 24 | 37 |  40 | 55 | 63 | 65 | 69 | 83 | 97 | 105 | 106 | 110 | 114 | 118 | 122 | Mean |
> | --- | --- | --- | --- | --- | --- | --- | --- | --- | --- | --- | --- | --- | --- | --- | --- | --- |
> VisMVSNet [3] | 0.98 | 2.10 | 0.93 | 0.46 | 1.89 | 0.67 | 0.67 | 1.08 | 0.67 | 0.96 | 0.66 | 0.85 | 0.30 | 0.45 | 0.51 | 0.88 |
> Ours | 0.66 | 1.01 | 0.71 | 0.43 | 1.06 | 0.99 | 0.73 | 1.43 | 1.18 | 0.78 | 0.64 | 0.93 | 0.38 | 0.54 | 0.54 | 0.80 |
> Ours (ft) | 0.55 | 0.71 | 0.39 | 0.38 | 0.79 | 0.65 | 0.67 | 1.29 | 0.96 | 0.64 | 0.49 | 0.59 | 0.33 | 0.44 | 0.45 | 0.62 |
>
> **Q3: The reconstruction without finetuning in Fig.8 is quite noisy**
>
> 1) Nosie is unavoidable in the reconstruction of generalizable model, and often requires some post-processing to filter, for example, photometric consistency and geometric consistency are required to filter in multi-view stereo.
> 2) Inferring accurate geometry from only three images is challenging, and the model is prone to ambiguity due to missing multi-view correspondence information. Even though, with our designed modules, we can recover more details than SparseNeuS, such as the eyes, mouth, and nose.
>
> **Q4: Does using position as input to MLP affect the generalization of the model?**
>
> This is like the positional encoding in the transformer. Our experiments show that it will not affect the generalization of the model, and it will speed up the convergence of the model, which has also been proved by SparseNeuS.
>
> **Q5: Does it make sense to blend the colors of all view projection? What if the point is not visible in certain view?**
>
> This is a crucial problem we faced during our implementation. From Eq. (3), we can see that the final color is blended as the weighted sum of the colors projected from the source views. The network is trained to estimate higher weights for those visible views and lower weights for others. Theoretically, it will make sense to blend the colors of all view projection as long as there is at least one visible view. However, as your concern, the existing of the occlusion will inevitably degrade the performance of the model. To mitigate this drawback, We adopt a robust training strategy that randomly selects N source views from the pool with both adjacent and non-adjacent views. In this way, the model is more robust to blend accurate color from situations with occluded views.
>
> **Q6: How to handle occlusion cases in the multi-scale feature consistency?**
>
> As you say, the features are naturally inconsistent if there are occlusions, and such bad cases will pollute the model training if no operation is done. To solve this problem, we only select the top K views with the largest feature consistency (the smallest NCC loss) to calculate the consistency loss, which is an effective empirical strategy assuming that the consistency with occluded views is small.
>
> **Reference**:
>
> [1] VolRecon: Volume Rendering of Signed Ray Distance Functions for Generalizable Multi-View Reconstruction, CVPR 2023.
>
> [2] Rethinking Rendering in Generalizable Neural Surface Reconstruction: A Learning-based Solution, arxiv 2023.
>
> [3] Visibility-aware Multi-view Stereo Network, BMVC 2020.

---

### Author Rebuttal · Authors · 2023-08-08

We sincerely thank all reviewers for their detail and constructive comments. We are glad that the reviewers appreciate the novelty (Tu79, EdqW, cro4), state-of-the-art and impressive results (cro4), reasonable ablation studies (EdqW), good technical soundness (cro4, XxbL), clear writing (XxbL). We highlight and summarize the advantages of our model here.

1. We are the first to successfully build a generalizable model that can reconstruct geometries comparable to or even superior than recent per-scene overfitting methods like NeuS and VolSDF. Meanwhile, we can achieve state-of-the-art performance with fast per-scene finetuning. Compared with existing generalizable model SparseNeuS, our model can achieve more than 60% improvement in the case of dense input (from 2.08 to 0.80), and more than 20% improvement in the case of sparse input (from 1.27 to 0.99). Moreover, our model can be trained end-to-end, while SparseNeuS needs two training stages.
|  | 24 | 37 |  40 | 55 | 63 | 65 | 69 | 83 | 97 | 105 | 106 | 110 | 114 | 118 | 122 | Mean |
| --- | --- | --- | --- | --- | --- | --- | --- | --- | --- | --- | --- | --- | --- | --- | --- | --- |
| SparseNeuS ft (Sparse) | 1.29 | 2.27|1.57|0.88|1.61|1.86|1.06|1.27|1.42|1.07|0.99|0.87|0.54|1.15|1.18|1.27|
| Ours ft (Sparse) | 0.75|1.99|1.34|0.75|1.12|1.68|0.77|1.21|0.98|0.74|0.89|0.54|0.43|0.82|0.91|0.99|
| SparseNeuS (dense) | 2.41|3.11|2.14|1.22|2.38|2.80|1.70|2.64|2.15|1.68|1.52|2.80|1.04|1.91|1.72|2.08|
| Ours (dense) | 0.66|1.01|0.71|0.43|1.06|0.99|0.73|1.43|1.18|0.78|0.64|0.93|0.38|0.54|0.54|0.80|

2. We are the first to introduce the multi-scale volume to the generalizable neural implicit surface reconstruction model to extend the single-volume model SparseNeuS. We found that increasing the number of input images leads to more severe occlusions and more noise, making geometric reconstruction more challenging. (This phenomenon has been proved in learning-based MVS methods (PVSNet)). We therefore introduce the low-resolution volume to maintain the globally coherent shape through the large receptive field, and adopt the high-resolution volume to recover finer detail. Meanwhile, each volume only equipped with a thin feature, which is more efficient than a single heavy high-resolution volume.
3. We introduced a self-enhanced multi-scale feature-metric consistency to improve the geometric consistency across multiple views. On the one hand, it can provide more accurate matching through the more discriminative multi-scale feature space and more robust patch similarity measure. On the other hand, the feature space can be self-enhanced during the training period, which will be continuously improved as the model converges.
4. We observe that different regions of the model have different numbers of visible viewpoints. The aggregated features of regions visible to a few viewpoints are likely to be polluted by irrelevant rays, making them less predictable. We thus design a view contrast loss to improve the model's reconstruction accuracy for these regions, thereby improving reconstruction smoothness.

The main concerns of all reviewers are concentrated on including more comparisons with existing methods. Here we have conducted more quantitative and qualitative experiments to further verify our effectiveness, and we will add these experiments to our final version. We put some experimental results in each response and put some in the uploaded PDF.

---

### Decision · Program_Chairs · 2023-09-21

**Decision:**

Accept (poster)

**Comment:**

This is a technically solid paper that received unanimous accept recommendations. The AC concurs and feels that the presented advances in generalizable sparse-view 3D inference approach would be beneficial to the community.